# Vessel Velocity Estimation and Docking Analysis: A Computer Vision Approach

João V. R. de Andrade [1,*,†], Bruno J. T. Fernandes [1,*,†], André R. L. C. Izídio [2], Nilson M. da Silva Filho [2] and Francisco Cruz [3,4]

1    Escola Politécnica de Pernambuco, University of Pernambuco, Recife 50720-001, Brazil
2    Suape, Complexo Industrial Portuário Governador Eraldo Gueiros, Ipojuca 55590-000, Brazil; andre.rafael@suape.pe.gov.br (A.R.L.C.I.); nilson.monteiro@suape.pe.gov.br (N.M.d.S.F.)
3    School of Computer Science and Engineering, University of New South Wales, Sydney, NSW 2052, Australia; f.cruz@unsw.edu.au
4    Escuela de Ingeniería, Universidad Central de Chile, Santiago 8330601, Chile
*    Correspondence: jvlc@ecomp.poli.br (J.V.R.d.A.); bjtf@ecomp.poli.br (B.J.T.F.)
†    These authors contributed equally to this work.

**Abstract:** The opportunities for leveraging technology to enhance the efficiency of vessel port activities are vast. Applying video analytics to model and optimize certain processes offers a remarkable way to improve overall operations. Within the realm of vessel port activities, two crucial processes are vessel approximation and the docking process. This work specifically focuses on developing a vessel velocity estimation model and a docking mooring analytical system using a computer vision approach. The study introduces algorithms for speed estimation and mooring bitt detection, leveraging techniques such as the Structural Similarity Index (SSIM) for precise image comparison. The obtained results highlight the effectiveness of the proposed algorithms, demonstrating satisfactory speed estimation capabilities and successful identification of tied cables on the mooring bitts. These advancements pave the way for enhanced safety and efficiency in vessel docking procedures. However, further research and improvements are necessary to address challenges related to occlusions and illumination variations and explore additional techniques to enhance the models' performance and applicability in real-world scenarios.

**Keywords:** vessel velocity estimation; docking analysis; computer vision; Structural Similarity Index (SSIM)

## 1. Introduction

The Suape complex in Pernambuco is considered a complete hub for industrial and port businesses in the Brazilian Northeast. Since its creation, the complex has attracted many companies interested in placing their products in the regional market or exporting them to other countries. Moreover, the privileged geographical position of Pernambuco, in the center of the Northeast Region, makes Suape a concentrator and distributor of goods, turning it into an international port (hub port) for all of South America since its external and internal ports offer the necessary conditions to serve large ships.

As a well-established port in the Brazilian scenario, the opportunities for using technology to make activities more efficient are immense. Video analytics can be used for some of the port's processes among these technologies. Video analytics are the automatic processing and understanding of video content to determine or detect spatiotemporal events and extract information or knowledge about the observed scene, utilizing a single camera or multiple cameras [1]. Using video analytics in a surveillance system in the port can bring greater productivity to the processes. Surveillance systems can play an essential role in vessel port processes by providing tools for situational awareness and monitoring maritime security [2].

In this context, the anchoring process can be modeled through video analytics. Every ship docking structure in a port has a capacity limit to absorb the mechanical energy transported by the docking vessel. An important factor during the docking of vessels is the approach velocity. Designing an analytics tool that measures docking velocity is crucial for the port, as approach speeds above the limit can cause damage to the port's infrastructure [3]. In addition, ship behavior prediction is critical for early detection of potentially risky behavior, such as ship collision. Such a model could also improve maritime traffic efficiency [4]. Currently, in SUAPE, such information is not available at all docking berths, and when available, it comes from more expensive third-party technologies, such as lasers.

Docking berths have structures called mooring bollards. The mooring bollards are structures where cables that hold the ships are tied. The mooring of these cables is used to determine the exact time of docking and undocking of the ships. Therefore, identifying the exact moment of docking of the ships is of utmost importance for port administration to know the vessel's time of stay. Currently, in SUAPE, port management relies on the labor of employees to identify the docking and undocking. It is important to note that in several ports, the criteria of docking and undocking are established based on subjective observations of operators and shipmasters, without particularized studies to support them [5].

Designing video analytics for these two problems has an impact on both infrastructure and revenue. From an infrastructure perspective, the Port Authority's interest in preserving the equipment involved in mooring processes becomes relevant. From a revenue perspective, it begins to be calculated when the first cable is tied to the bollard during mooring and ends when the last cable is removed during unmooring. In addition, the information from this monitoring will provide the Port Authority with subsidies for determining and assigning responsibilities in the case of incidents and damage to the equipment involved in the mooring process.

This work aims to utilize two computer vision techniques to address these problems. The border-following algorithm is employed in the velocity estimation model, while the Structural Similarity Index Measure (SSIM) is utilized to estimate the time of vessel docking and undocking. All experiments presented in this paper were conducted using a dataset provided by the port authorities of SUAPE. However, it is important to note that both systems are still in the initial phase of implementation, and there is a lack of ground truth data to compare our velocity estimation results. Additionally, the provided dataset has limited diversity in terms of scenarios. Despite these limitations, the preliminary results obtained were deemed satisfactory by the port supervisor.

## 2. Theoretical Foundation

This section presents the two main theoretical concepts utilized in the proposed systems. First, this paper's vessel speed estimation strategy is based on the border-following algorithm originally proposed by Suzuki and Abe [6]. In addition, the detection strategy for ropes on the bollards utilizes the Structural Similarity Index Measure (SSIM) [7] to compare a target image with an input image. These methodologies form the foundation for the suggested systems.

Incorporating well-established theoretical concepts into the systems enhances the effectiveness of velocity estimation and tied rope detection. These methodologies play a crucial role in improving the efficiency and safety of port operations, facilitating smoother vessel approximations and successful docking procedures. Integrating these proven approaches enables more accurate estimations and contributes to the overall optimization of port activities.

### 2.1. Border Following

An image is not simply a random and insignificant collection of pixels but represents a meaningful arrangement of regions and objects. The human brain has the remarkable

ability to organize these pixels into something that makes sense in the human world, even in the presence of noise. Previous research has identified various factors, such as similarity, proximity, continuity, symmetry, parallelism, closeness, and familiarity, as the key elements influencing the human perception of an image. Understanding and modeling these factors are crucial in capturing the perceptual grouping process [8].

One strategy commonly used to capture the perceptual grouping process is contour detection. Quantifying the presence of boundaries at specific locations in an image through local measurements is a widely employed approach in contour detection. These local measurements can provide valuable information about the presence and strength of boundaries, enabling the identification and segmentation of objects in the image. Various techniques and algorithms have been developed to enhance contour detection and improve the accuracy of boundary localization [9].

Accurately identifying boundary localization plays a crucial role in border following, a fundamental technique in processing digitized binary images. Border following has been extensively studied due to its wide range of applications, such as picture recognition, picture analysis, and image data compression [6]. Furthermore, by accurately tracing the boundaries of objects within an image, border-following techniques contribute to various computer vision tasks, enabling efficient and reliable analysis and interpretation of visual data.

### 2.2. Structural Similarity Index Measure

Digital images can exhibit various types of distortions. Therefore, in many applications where images are intended for human viewing, human subjective evaluation is considered the most accurate approach [7]. However, this subjective evaluation method is time-consuming and requires the involvement of multiple observers [10]. In response to these limitations, researchers have proposed objective approaches that can automatically estimate the visual quality of images without the need for human intervention [11]. These objective methods aim to provide efficient and reliable metrics for image quality assessment.

One of the earliest objective quality metrics used to assess image quality is the Peak Signal-to-Noise Ratio (PSNR) between a base image and a compared image [12]. The PSNR metric is commonly used in image processing literature as a converted Mean Squared Error (MSE) measure. While the PSNR/MSE metric is widely employed and has historically been used to optimize various signal processing applications, it does not always align with human perception. For example, experiments have shown that images with different types of noise can have the same PSNR value, despite one being visually superior based on human perception [13].

The limitations of the PSNR metric have led researchers to propose alternative objective quality metrics that aim to better align with human perception and capture the visual quality of images more accurately [14]. The Structural Similarity Index (SSIM) is an objective quality metric used in image processing. It quantifies the similarity between two images by considering their structural information, luminance, and contrast. The SSIM considers the visual system's characteristics and perception, which makes it more reliable in assessing image quality than traditional metrics [7].

Equation (1) shows the SSIM equation, where $x$ and $y$ are the input images that are being compared, $u_i$ are the means, $C_1$ and $C_2$ are constants added to avoid division by zero, $\sigma_i^2$ are the variances, and $\sigma_{xy}$ is the covariance between the two compared images.

$$\text{SSIM}(x,y) = \frac{(2\mu_x\mu_y + C_1)(2\sigma_{xy} + C_2)}{(\mu_x^2 + \mu_y^2 + C_1)(\sigma_x^2 + \sigma_y^2 + C_2)} \tag{1}$$

The SSIM scores are more coherent than other MSE scores when considering human visual perception. It is observed that degradation in intensity and contrast does not significantly degrade the image structure. On the other hand, noise contamination is considered a process that affects the image structure. Developing a metric that considers these factors helps construct models that closely align with human perception [13].

## 3. Related Works

Zwemer et al. [15] proposed a system that uses two cameras to estimate the velocity of ships. In addition, the proposed system detects and tracks ships through re-identification (re-ID) tasks. The re-ID task is used to link ships seen by the two cameras. The authors introduced a new Vessel-reID dataset to train and evaluate the model. In this dataset, the authors evaluated various machine learning models. They found that the SSD multibox detector model can cover a large portion of the total trajectory to re-identify vessels through successful detections.

The authors' system consists of two stages: camera processing and vessel re-identification. The camera processing stage detects and tracks vessels in real time, storing crops and a timestamp in the database to create a unique feature vector description. The second stage evaluates visually matching vessels using travel time constraints and generates an alert if a match is found for a vessel that traveled faster than the speed limit.

The authors highlight the challenges identified, indicating that measuring the velocity of a vessel using only one camera is affected by wind interference, causing small camera movements, difficulty in establishing the exact position of the vessel, and weather and lighting conditions. Hence, measuring during larger displacements is recommended. In addition, different types of vessels are also a challenge, including those that occupy the entire camera image or small rowing vessels that occupy small pixels in the image.

Wawrzyniak et al. [16] presented a method for detecting and tracking ships. The method consists of a Moving Vessel Detection Algorithm (MVDA), Status Update Algorithm (SUA), and Temporary Buffer (TB). The MVDA returns the probable bounding boxes of ships in the scene given a video input. The results are stored in the TB at the end of each prediction round, and the SUA analyzes the contents of the TB, filters probable artifacts, and returns zero or more images of moving ships.

In addition to the vessel detection and tracking algorithms, the authors applied a water detection algorithm to help the model discard water features behind ships. The water detection algorithm used the contours of objects, and for each contour, it calculated the length and area occupied. If the length or area of the contour was smaller than a previously established threshold, that region was labeled "water". This additional step was important because water can often be mistaken for a ship's hull, leading to false positives and inaccurate tracking. This is particularly important in scenarios where multiple vessels are present, and there is a lot of movement in the water, such as in busy ports or shipping lanes.

The authors, like Zwemer et al. [15], have identified that incorrect detections frequently occur due to unfavorable lighting conditions. In addition, since some evaluated scenarios come from cameras with less color saturation, the background subtraction algorithm performs worse in these scenarios.

Regarding the docking problem, Alvarellos et al. [5] presented a new computer vision technique for monitoring docked ships. The proposed technique uses visual feature correlation of ship images to estimate its movement over time. The similarity metric used, Pearson similarity, according to the authors, produced a robust model to handle changes in illumination, allowing the model to perform well in various scenarios. During the tests, this approach had promising results. The heave motion recorded with the optical method showed a very satisfactory correspondence with the wave height measured by Radar.

Dosh and Yilmaz [17] utilized the structural similarity (SSIM) measure [7] for detecting anomalies. SSIM is a widely used image quality assessment method that compares the structural similarity between two images, considering luminance, contrast, and structural information. This method has been shown to perform well in detecting changes in complex scenes, such as traffic videos. Furthermore, the authors suggest that using SSIM as a measure of similarity allows for more accurate detection of anomalies, such as stalled vehicles, as the similarity concerning an image with a stalled vehicle would remain close to zero and increase significantly when a vehicle appears.

## 4. Methodology

The methodology employed in this study aims to address the key objectives of vessel speed estimation and mooring analysis using a computer vision approach. The following subsections outline the steps and techniques involved in each process.

### 4.1. Vessel Velocity Estimation Strategy

The strategy for measuring vessel velocity comprises two stages: processing and measurement. In the processing stage, image purity is enhanced to identify ship features accurately. The Suzuki and Abe algorithm [6] is applied to detect image borders. In contrast, a water feature detection algorithm inspired by [16] was utilized to identify water areas without compromising ship element edges. To facilitate the detection of these features, the following sequence of image processing steps was applied to the images:

- Conversion to grayscale. The original RGB image was converted to a single color channel, varying its intensity between 0 (black) and 255 (white). Intermediate values represent shades of gray.
- Binarization. With the image in grayscale, a threshold intensity value, or thresh, was established to determine whether that pixel would be considered black or white, depending on its intensity. In the case described here, it was established that all pixels with intensity below 150 would be considered black and those above, consequently, white.

After these steps, the water detection algorithm separated the ship's edge from the water. Once the ship's edge could be isolated from the water movement, identifying the approach velocity became a relatively simple task. Based on the concept that average velocity is defined as the ratio between the variation in space and the variation in time, speed estimation utilized two different points. When the vessel executes a translation movement associated with a rotation movement, as often happens during docking processes, the vessel does not always approach parallel to the dock. Therefore, it was relevant to measure the velocity at two distinct points: on the left side of the image and the right side of it.

Thus, two detection zones were created on the left side, one further away and one closer to the dock, aligned vertically and 5 m apart. The same was done on the right side. Each detection zone monitored the change in pixels so that the time of that event was recorded from a certain number of altered pixels. Therefore, when the first detection zone (farther from the dock) perceives the vessel's presence, the time starts to be counted. The counting is finished when the second detection zone (closer to the dock) is activated; thus, the average velocity is calculated.

The conversion of pixel velocity to real-world velocity was achieved through a simple image-based coordinate mapping technique known as homography. During the data collection process, the port authorities provided real-world coordinate information by utilizing reference points in the upper and bottom detection zones on the left and right sides. These reference points were then mapped onto the image plane. By establishing a proportional relationship between the image coordinates and the measurements provided by the port authorities, the algorithm could convert the image coordinates into real-world coordinates and estimate the velocity. Since the portion of the vessel considered for calculating the speed moves in a planar environment, such as the sea, perspective distortion has a negligible effect on the calculation, thus enhancing the accuracy of the estimation. The applied homography transformation follows the following equation

$$\begin{bmatrix} x' \\ y' \\ 1 \end{bmatrix} = \begin{bmatrix} h_{11} & h_{12} & h_{13} \\ h_{21} & h_{22} & h_{23} \\ h_{31} & h_{32} & h_{33} \end{bmatrix} \begin{bmatrix} x \\ y \\ 1 \end{bmatrix} \tag{2}$$

where $x'$ and $y'$ represent the real-world coordinates, and $x$ and $y$ correspond to the image coordinates. The $h_{ij}$ parameters are components of the projection matrix $H$, which is

calculated using the Singular Value Decomposition (SVD) technique, taking into account the coordinates provided by the port authorities.

The approximation angle is calculated using the same projection matrix. The centroid of the contour within the detection zones is considered. Specifically, the centroid of the contour in the left detection zone is compared with that of the right zone contour. The approximation angle is then calculated using triangular trigonometric relations between these points. This approximation angle provides valuable information about the orientation or alignment of the detected object. Figure 1 illustrates the velocity estimation model diagram.

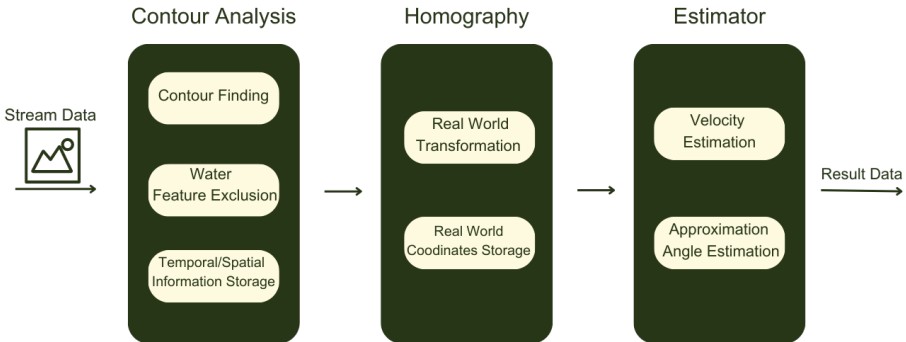

**Figure 1.** Vessel velocity model diagram.

### 4.2. Ropes on the Bollards Detection

To analyze the docking and undocking times, one strategy has been developed for detecting the ropes on the mooring bitts. According to the information provided by the port authorities, a vessel is considered "docked" when it is securely tied to 2 to 8 mooring bitts. Once all the required mooring bitts were securely tied, the docking time was recorded. It is important to note that the synchronization of each mooring bitt detection was not part of our system development. The responsibility for synchronizing the mooring bitts lies with the port employees. Our work primarily focuses on the classification algorithm for the mooring bitts. This subsection will outline the strategy employed to analyze each individual mooring bitt.

The strategy used for identifying the cable tied to the mooring bitt was the comparison of image similarity through the Structural Similarity Index (SSIM). SSIM [7] is a perception metric that quantifies image quality degradation caused by processing such as data compression or loss during data transmission. It is a complete reference metric that requires two images from the same image capture, a reference image, and a posterior image. Thus, an image with the cable tied to the bitt was used as a reference to identify whether the cable was tied. Prior to using SSIM, a sequence of treatments was implemented to reduce noise and, primarily, the effects of illumination.

The treatments were as follows:

- Definition of the analysis region: The image was initially cropped to define the Region of Interest (ROI), which in this case, was the limits of the mooring bitt.
- Removal of the background: The image background was removed to avoid possible distortions caused by features unrelated to the ROI.
- Conversion to grayscale: The original RGB image then only had one color channel, varying in intensity between 0 (black) and 255 (white).
- Histogram equalization: The image's histogram was equalized, which aims to improve contrast, to expand the intensity range.
- Binarization: Being in grayscale, a threshold intensity value, or thresh, was established to determine whether that pixel should be considered or not.

After the image processing steps were applied, the image with the cable tied to the mooring bollard was used as the reference image for comparison. All subsequent input images were compared to this reference image using the Structural Similarity Index (SSIM)

metric. A grid search was performed on the training dataset to determine the optimal SSIM threshold parameter ($\theta_1$). The search involved trying different threshold values and evaluating their performance in terms of accuracy and precision. The goal was to identify the best threshold value to classify tied and non-tied images accurately.

Images with SSIM values above $\theta_1$ were considered to be tied images, while images with SSIM values below $\theta_1$ were classified as non-tied images. This threshold was chosen to ensure a high level of accuracy in identifying images with the cable tied to the mooring bollard while minimizing false positives. The algorithm's performance was evaluated using a separate test dataset, and the results showed that the algorithm could accurately identify tied images with a high degree of precision. The Figure 2 shows the flowchart of proposed strategy.

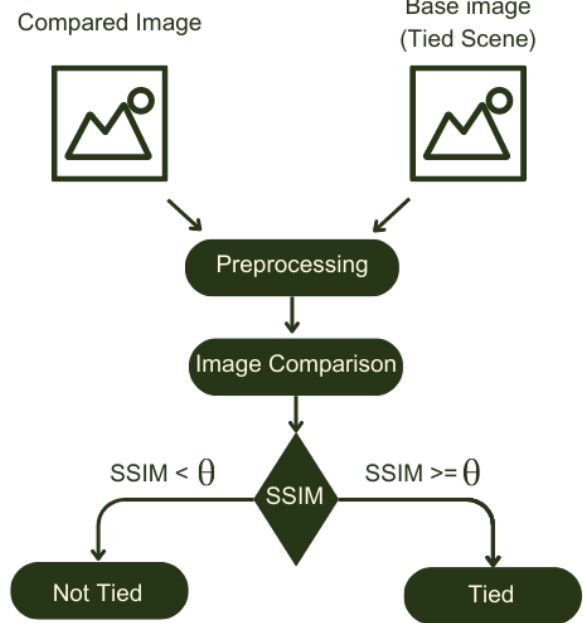

**Figure 2.** Ropes in the mooring bitt model.

## 5. Experimental Dataset

The dataset used in this project was mainly produced at Berth 5 of the Suape Port in Brazil. The videos were recorded using mobile phones with HD resolution at 30 FPS, positioned on a tripod 2 m to the sea and 1.5 m above the ground, always in front of the mooring bollards. The predominant weather during the recordings was sunny, with good lighting and visibility. The objective of this positioning was to obtain the speed and angle of approach of the vessel and verify its mooring condition, which coincides with the concepts of "berthed" or "unberthed" of the ship in relation to the berth. Figure 3 shows the dataset scenario.

Once the data were collected, the video selection process began, with the criterion of selecting those with better image clarity and less vibration due to wind action on the camera. Next, five berthing maneuvers were filmed from two perspectives (two cameras positioned at different mooring bollards, recording simultaneously). The selected videos were then used to develop and evaluate the proposed algorithms, which aimed to detect and track vessels in real time and monitor their mooring status.

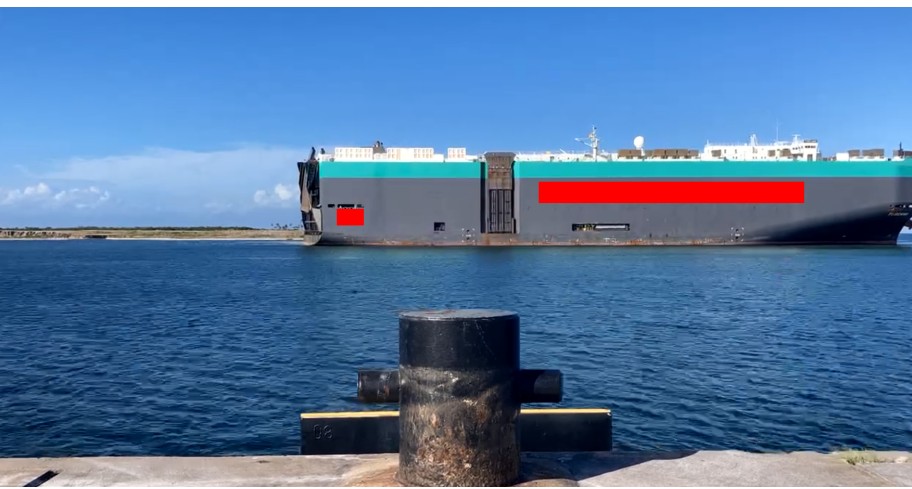

**Figure 3.** Dataset scenario.

Although the data were collected in sunny weather conditions without any adverse climate changes, there was an occlusion problem in the dataset that directly impacted the classification of "tied" and "not tied" ropes on the mooring bitt. The images provided by the port authorities sometimes contained occlusion issues, as employees at the port often obstruct the view of the mooring bitt. The presence of obstructing objects or individuals can cause partial or complete occlusion, leading to incorrect classification results. Figure 4 illustrates an example of occlusion in the dataset.

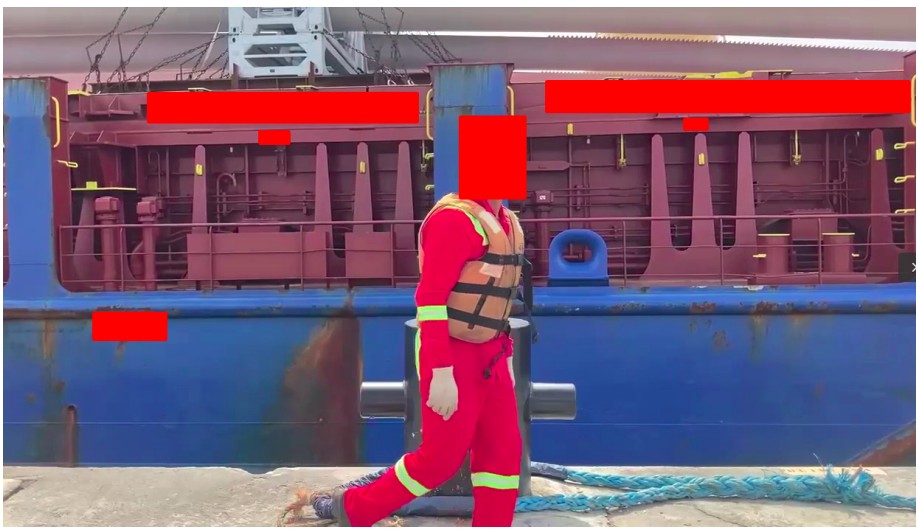

**Figure 4.** Occlusion example.

## 6. Experimental Results

The vessel speed estimation was performed using several videos of the vessel approaches from the dataset. First, the approach was divided into left and right sides, and for each side, two detection zones were created to measure the vessel's velocity. The detection zones were represented by white rectangles in Figure 5, and the vessel contours were perceived within these zones. Next, the two detection zones for each side were placed vertically, with one zone being farther away from the berth than the other, with a distance of 5 m between them.

Table 1 presents the estimated velocity and approximation angle obtained through the algorithm. Velocity was measured in millimeters per second, providing insights into the speed of vessel movement, while the angle, measured in degrees, allowed for an understanding of the approach trajectory.

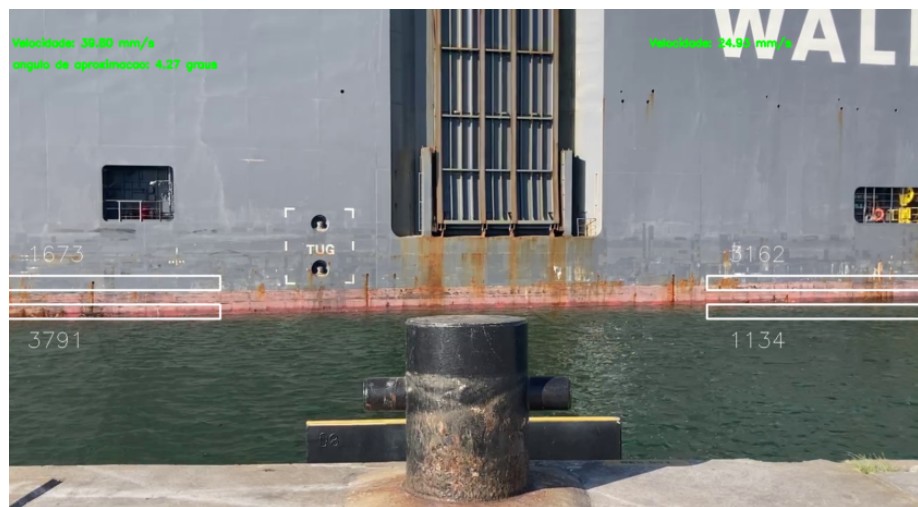

**Figure 5.** Velocity and angle detection screen.

**Table 1.** Velocity and angle estimated by the algorithm.

| Scene | Left Velocity (mm/s) | Right Velocity (mm/s) | Approximation Angle (degrees) |
|---|---|---|---|
| Scene 1 | 45.33 | 27.39 | 4.52 |
| Scene 2 | 45.35 | 27.42 | 4.52 |
| Scene 3 | 45.33 | 27.39 | 4.51 |
| Scene 4 | 45.19 | 27.35 | 4.52 |
| Scene 5 | 45.34 | 27.40 | 4.52 |

The region of interest (ROI) used for the experiments was defined as the boundaries of the mooring bitt. Figure 6 illustrates the ROI employed in the analysis. By focusing on this specific area, the system could extract relevant information and accurately assess the algorithm's performance.

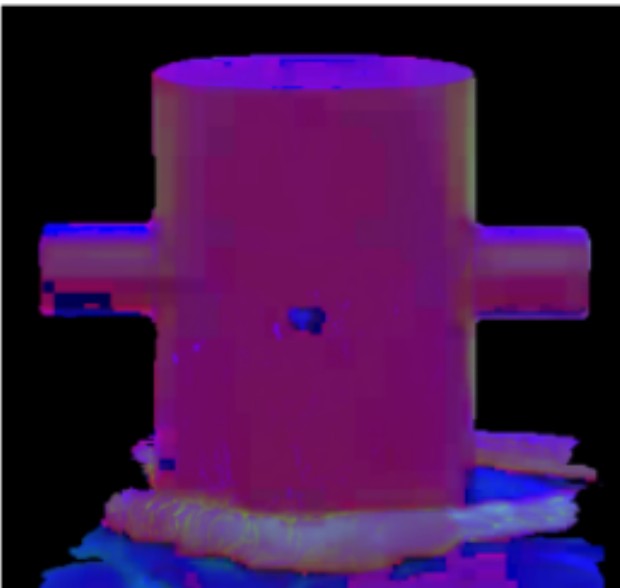

**Figure 6.** Mooring ROI.

To evaluate the SSIM strategy for identifying the presence of a tied cable on the mooring bitt, which serves as the initial trigger for the docking process, a total of 220 images were labeled as "tied" or "not tied". The optimal SSIM threshold parameter ($\theta_1$) was

obtained through grid search. The optimal value found in the training set was 0.81. For a base image labeled as "tied", SSIM values measured above $\theta_1$ were considered "tied", while values below $\theta_1$ were considered "not tied". Each class's precision and recall values are presented in Table 2.

**Table 2.** Precision and Recall of the SSIM strategy.

| Class | Precision | Recall |
|---|---|---|
| Tied | 1.00 | 0.62 |
| Not Tied | 0.78 | 1.00 |

As observed in Table 2, the recall of the "tied" class and the precision of the "not tied" class were relatively low. However, this issue arises primarily due to occlusions present in the dataset. As discussed in Section 5, the dataset contains images with occlusions caused by port employees. These occluded images significantly differ from the base image labeled as "tied," resulting in their frequent misclassification as "not tied." Consequently, the precision of the "not tied" class is reduced, and the recall of the "tied" class is also affected.

## 7. Limitations and Further Research

Although the results of the speed method were considered good by the port authorities, the speed estimation model results were obtained by analyzing videos, where the gap between the upper and lower regions was measured on-site. Unfortunately, there was a lack of other velocity measurement systems in the region where the data were collected. However, during the implementation of the system, the port would have a ground truth system in place. The radar and GPS data of the vessels would be provided by the port authorities, and the system would utilize this data to validate the velocity estimated by our computer vision approach. Subsequently, this will leverage this data source to make refinements to the model and enhance its errors. This will allow for the comparison of the developed model with other baseline methods and provide valuable insights into the model's effectiveness and potential areas for further improvement.

The algorithm effectively identified the presence of tied cables on the mooring bitts, demonstrating its capability in the docking process. However, it is important to note that certain challenges persisted despite the removal of images with high variations in illumination and saturation from the experimental dataset, as described in Section 5. Some images still exhibited occlusions and illumination issues, which impacted the performance of the algorithm. These challenging images led to a decrease in the recall of the "tied" class, highlighting the need for further refinements. Further research entails the advancement of robust algorithms capable of effectively dealing with occlusions and enhancing the accuracy of classification, particularly in real-world scenarios pertaining to mooring bitts. Additionally, future studies may involve analyzing the algorithm's performance with different types of ropes made from various materials and assessing the accuracy of classification for each type. Moreover, incorporating a phase to classify rope tension can be applied to the model to detect incorrect ties on the mooring bitts. The application of concepts such as Siamese Networks, harnessing the power of Artificial Neural Networks, holds promise for improving the metrics of the developed system.

Another limitation of the mooring bit analysis arises from the utilization of SSIM. Results obtained by [18] demonstrate that SSIM may not always accurately determine the similarity between images of the same scene. The authors further suggest that Pearson linear correlation could serve as a more suitable tool for similarity analysis. Therefore, future research can explore alternative metrics such as Pearson correlation coefficient to overcome this limitation.

## 8. Conclusions

Both results were classified as good by the port authorities of SUAPE. The camera positioning, which allowed for the utilization of two analytics simultaneously, was a key factor in the implementation of the system at the port.

The results of the vessel speed estimation demonstrated the effectiveness of the algorithm in accurately measuring the velocity of approaching vessels. The algorithm was able to capture the dynamic changes in velocity, allowing for precise analysis and monitoring of the vessels' movements. The obtained results not only showcased the algorithm's ability to estimate vessel speed but also highlighted its potential applications in optimizing docking procedures and ensuring safe and efficient maritime operations. These findings contribute to advancing computer vision techniques in the maritime industry, offering valuable tools for enhanced vessel management and navigation.

As future work in vessel velocity estimation, refining the algorithms used for motion detection and tracking can improve the model. In addition, other developments, such as exploring advanced computer vision techniques and considering integrating other data sources such as radar or GPS complement the visual analysis. In the initial implementation of our vision-based approach, GPS and radar data will be used as evaluation techniques. The utilization of these devices provides a satisfactory approximation of the velocity. However, in further development, once the precision of the velocity model reaches a mature level, it becomes feasible to rely solely on the camera-based system for velocity estimation. This transition is motivated by the lower maintenance cost of the camera system compared to GPS and radar, making it a more cost-effective solution for long-term operation at the pier. Additionally, incorporating machine learning approaches to learn and adapt to different vessel types and environmental conditions can further enhance speed estimation accuracy.

The ropes on the bollards' algorithm presented in this study demonstrate computer vision techniques' utilization in analyzing and monitoring vessel docking processes. By employing advanced algorithms, such as the structural similarity index (SSIM), the algorithm identifies the presence of a tied cable on the mooring bit. However, as discussed in Section 7, certain limitations have been identified. These limitations can be addressed through the utilization of alternative metrics, such as Pearson correlation, and the exploration of techniques such as neural Siamese networks, which offer the potential for further improvement.

Despite the identified limitations, the algorithm developed for analyzing the ropes on the bollards can play a significant role in initiating the docking process. Its implementation offers promising prospects for integration into real-world maritime operations. By automating the detection of tied cables, the algorithm can enhance situational awareness, improve docking efficiency, and enhance overall safety in port operations. Moreover, the algorithm's adaptability and scalability make it applicable to various docking scenarios and vessel types, further expanding its potential impact.

While the current model focuses on detecting tied cables on the mooring bit, future works could expand the algorithm to include the analysis of other components involved in the mooring process, such as fenders or bollards. Another potential direction is to incorporate the evaluation of rope position in the mooring bitt. This inclusion can provide a more comprehensive understanding of the docking dynamics. Additionally, investigating the use of advanced image processing techniques to detect and analyze the tension in the mooring lines can contribute to assessing the stability and safety of the moored vessel.

**Author Contributions:** Conceptualization, B.J.T.F., A.R.L.C.I. and N.M.d.S.F.; methodology, J.V.R.d.A., B.J.T.F., A.R.L.C.I. and N.M.d.S.F.; software, J.V.R.d.A., A.R.L.C.I. and N.M.d.S.F.; validation, J.V.R.d.A., A.R.L.C.I. and N.M.d.S.F.; formal analysis, J.V.R.d.A., B.J.T.F., A.R.L.C.I., N.M.d.S.F. and F.C.; investigation, J.V.R.d.A., B.J.T.F., A.R.L.C.I., N.M.d.S.F. and F.C.; resources, B.J.T.F., N.M.d.S.F. and F.C.; data curation, J.V.R.d.A., A.R.L.C.I. and N.M.d.S.F.; writing—original draft preparation, J.V.R.d.A., A.R.L.C.I. and N.M.d.S.F.; writing—review and editing, J.V.R.d.A., B.J.T.F., and F.C.; visualization, J.V.R.d.A., B.J.T.F., A.R.L.C.I., N.M.d.S.F. and F.C.; supervision, B.J.T.F.; project administration, B.J.T.F.;

funding acquisition, B.J.T.F. and F.C. All authors have read and agreed to the published version of the manuscript.

**Funding:** This research was funded by Fundação de Amparo à Ciência e Tecnologia de Pernambuco (FACEPE) grants numbers APQ-0356-1.03/21. and APQ-0072-1.03/22.

**Data Availability Statement:** Data available on request. The data presented in this study are available on request from the corresponding author. The data are not publicly available due to privacy concerns for the port's staff, who may be captured in the scenes.

**Acknowledgments:** The authors would like to acknowledge the support and resources provided by Projeta.AI, which greatly contributed to the development and implementation of the proposed methodologies. Their expertise and data played a significant role in the success of this research project.

**Conflicts of Interest:** The authors declare no conflict of interest.

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
