# Peer review of "Vessel Velocity Estimation and Docking Analysis: A Computer Vision Approach"

_algorithms, doi:10.3390/a16070326_

Round 1
Reviewer 1 Report
Dear authors,
thank you for submitting the paper. I have the following remarks:
Overall, the topic is quite interesting as a real-world study applying image detection and processing technology to a domain which is driven mostly manually all over the world. I encourage you to continue.
My major concern is, that at the current state, the working results are not convincing enough to work as a "proof of concept" nor as a schema that can be applied by others nor as a sound scientific contribution. I suggest to (i) work further on the "general applicability" of the technology stack you are developing (i.e. reduce the manual parts of the work) and (ii) on the convincing evaluation of the accuracy
e.g. there is no ground truth (yet) for the vessel velocity estimation. It is unclear how you convert from "pixel speeds" to real-world speeds. Data collection and filtering sounds quite biased as you only take good weather conditions into account, one location and 5 maneuvers etc.
Minor:
table i: units of the quantities?
Precision and recall of “tied” and “not tied”: with the parameters you have, why did you chose this “working point” on the ROC curve for your approach? a recall of 0.62 for Tied is quite low, given, that usually one can go down with the precision a bit and increase the recall. Same for the “not tied” class with “only” a precision of 0.78, but a recall of 1.0.
Not a native speaker, though I assess the level of English as good.
Author Response
Dear reviewer,
Thank you for your valuable feedback. We appreciate the revisions you suggested, which have greatly improved several sections of the article and provided us with new insights for future research in the same domain.
Reviewer – Data collection and filtering sounds quite biased as you only take good weather conditions into account, one location and 5 maneuvers etc.
We acknowledge the limitations of our dataset, which is biased towards clear weather conditions. The data was collected during the sunnier months of Ipojuca, where the port is located, and thus lacks representation of cloudy situations. However, based on discussions with the port authorities, focusing on sunny weather conditions was deemed acceptable for our initial approach. Rest assured, we are actively working to collect additional data that encompasses a broader range of weather conditions for future research.
Reviewer – There is no ground truth (yet) for the vessel velocity estimation. It is unclear how you convert from "pixel speeds" to real-world speeds
Regarding the absence of a ground truth system for the velocity model, it is important to note that the port did not possess such a system during the study. Consequently, we proceeded without a ground truth and presented the preliminary results to the port authorities. However, we recognize the significance of incorporating a ground truth system to validate our approach and make finer adjustments. We are currently in the process of collecting data with a dedicated ground truth system for velocity, which will enable us to validate our system more accurately.
Reviewer – Minor Questions
We have also revised various sections of the article and included additional figures that detail the strategies employed and their potential to enhance existing vessel port processes. Moreover, we have explained the factors contributing to the low recall and precision, specifically highlighting the occlusion issues frequently encountered in the captured videos.
Once again, we appreciate your valuable input and suggestions, which have significantly contributed to the refinement of our research.
Reviewer 2 Report
The paper discusses a computer vision approach to the ship docking problem. The presence of a cable on a bollard is determined using the structural similarity metric of two images (SSIM). The authors collected a sample of images (mainly in sunny weather), which included an empty bollard and a bollard with a rope thrown over it. Images with a rope thrown was used as references, and the degree of similarity between the new frame and the reference was characterized using SSIM.
The work should be evaluated positively, however, the choice of metric seems to be not entirely successful. In some studies, for example, [1,2], it is stated that using SSIM it is possible to recognize only visually very similar images, while when comparing very different scenes, the Pearson correlation coefficient shows the best accuracy.
As recommendations for further research, the authors can be advised:
1) consider Siamese neural networks designed to evaluate the similarity of two images;
2) diversify the training set with frames with conditions close to real: with bad weather, night shooting or overexposed images. For these cases, you may have to drop the SSIM metric.
Literature
1. Starovoitov V.V., Eldarova E.E., Iskakov K.T. Comparative analysis of the SSIM index and the Pearson coefficient as a criterion for image similarity// Eurasian journal of mathematicaland computer applicationsissn 2306–6172 volume 8, issue 1 (2020) 76 – 90
2. Avneet Kaur , Lakhwinder Kaur and Savita Gupta Image Recognition using Coefficient of Correlation and Structural SIMilarity Index in Uncontrolled Environment // International Journal of Computer Applications (0975 – 8887) Volume 59– No.5, December 2012
Author Response
Dear reviewer,
We sincerely appreciate your valuable feedback on our article. Your suggestions have played a significant role in enhancing the quality of our research and have provided us with fresh insights for further exploration in the same domain.
Reviewer - Pearson correlation coefficient shows the best accuracy [...] Consider Siamese neural networks designed to evaluate the similarity of two images;
Specifically, we would like to thank you for recommending the utilization of Siamese neural networks and the application of Pearson correlation in future approaches. Based on your revision, we have made some revisions to Section 7 of the article, which is now titled "Limitations and Further Research." In this section, we have included a brief paragraph addressing the conclusion of Starovoitov’s article, emphasizing the potential use of Pearson correlation in future approaches.
Reviewer 3 Report
The paper is well organized. The introduction is concise but sufficient to understand the scope of developments and application. The methods are also clearly illustrated.
With regard to the results, the reviewer believes that the real applicability of the procedure when the mooring lines can overlap each other around the bollards is not sufficiently documented. Also, the effects of the use of different materials (and therefore of sections) for the mooring lines have not been illustrated as well as that of the tensioning of the lines. In particular, even if the perfect functioning of the procedure is admitted in the presence of a single mooring line, it is not clear whether and how the recognition performances vary when the mooring conditions change. In conclusion, the illustrated procedure appears promising even if its practical applicability has not been demonstrated. Therefore, I suggest to further detail the applicability of the procedure to real ship mooring conditions.
The paper is written in clear and fluent English.
Author Response
Dear reviewer,
We sincerely appreciate your valuable feedback on our article. We have taken your suggestions into consideration and made several updates to the article.
Reviewer - In particular, even if the perfect functioning of the procedure is admitted in the presence of a single mooring line […]
In subsection 4.2, after your feedback, we have clarified that our model focuses on a single mooring bitt, while the port authorities handle the clustering of mooring bitts and their corresponding piers.
Reviewer - Also, the effects of the use of different materials (and therefore of sections) for the mooring lines have not been illustrated
Regarding the differentiation of rope types, we acknowledge that the current dataset consists of videos featuring the same material for the ropes used in mooring the vessels. However, we are actively working with the port authorities to collect additional data that includes different types of ropes to evaluate the algorithm's performance in varied scenarios.
Reviewer - As well as that of the tensioning of the lines
The current algorithm does not account for the tensioning of the lines. After the reviewer’s feedback, we have addressed this limitation in Section 7, where we discuss the algorithm's limitations and outline future research directions.
Reviewer - I suggest to further detail the applicability of the procedure to real ship mooring conditions.
We have also revised various sections of the article and included additional figures that detail the strategies employed and their potential to enhance existing vessel port processes.
Reviewer 4 Report
The paper "Vessel Velocity Estimation and Docking Analysis: A Computer Vision Approach" introduces a study that utilizes techniques, including edge detection and the Structural Similarity Index (SSIM), for speed estimation and mooring bitt detection. The evaluation conducted on the collected dataset validates the algorithms' efficacy.
But I have the following significant concerns,
(1) In the Conclusion section, the author suggests integrating other data sources such as radar or GPS to enhance the visual analysis and potentially achieve better results. Considering that GPS can provide accurate measurements for vessel velocity estimation and radar can utilize the Doppler effect for velocity estimation, incorporating these data sources could offer additional benefits. Moreover, setting the docking position as a specific GPS coordinate and using radar to track the ship's duration in certain areas can further improve the docking process. It would be valuable for the authors to validate and elaborate on these potential benefits to provide readers with a better understanding of the significance of their methods. If GPS and radar seem to be good solutions for their problem, why their methods are better?
(2) In Page 5, Section 4.1, the conclusion of the vessel velocity estimation is somewhat unclear. While Fig 2 displays the detection zones for the left and right sides, it is not evident how the pixel distance is translated into physical distance. It would greatly enhance the understanding if the authors included a figure illustrating their calculation process for converting pixel distance to physical distance.
(3) In the docking analysis, the comparison of SSIM between ship docked and undocked, as mentioned, raises some concerns about the reliability of this analysis. Factors such as lighting conditions or other visual changes might affect the results. It is crucial for the authors to provide evidence or conduct further experiments to demonstrate the effectiveness of their approach under varying visual conditions.
(4) In Page 7, Section 7, the author mentions the presence of a ground truth system during the implementation of the system. It would be beneficial to provide additional details about this ground truth system, such as how it is established or what it consists of. This clarification will help readers better understand the role and reliability of the ground truth system in the evaluation and validation of the proposed approach.
Generally, the Quality of English Language is good.
Author Response
Dear reviewer,
Thank you for your feedback. It has been instrumental in improving our article and providing us with new perspectives for further research.
Regarding your concerns:
Reviewer - […] 1 It would be valuable for the authors to validate and elaborate on these potential benefits to provide readers with a better understanding of the significance of their methods. If GPS and radar seem to be good solutions for their problem, why their methods are better? [...] 4 In Page 7, Section 7, the author mentions the presence of a ground truth system during the implementation of the system. It would be beneficial to provide additional details about this ground truth system, such as how it is established or what it consists of [...]
We would like to clarify that in the initial implementation of our computer vision approach, we will be utilizing radar and GPS as a ground truth system to validate the accuracy of our methodology. This will help us refine our computer vision algorithms. Once we have achieved the desired level of accuracy, we plan to transition solely to our computer vision approach as it is more cost-effective to maintain cameras compared to radar and GPS systems. After your review, we have included a brief paragraph in the article explaining it.
Reviewer - It is not evident how the pixel distance is translated into physical distance. It would greatly enhance the understanding if the authors included a figure illustrating their calculation process for converting pixel distance to physical distance.
We have incorporated more details about the velocity model explaining the model in subsection 4.1 of the article based on your feedback.
Reviewer - Factors such as lighting conditions or other visual changes might affect the results. It is crucial for the authors to provide evidence or conduct further experiments to demonstrate the effectiveness of their approach under varying visual conditions.
Unfortunately, our data collection period was limited to the clearer months of Ipojuca, where the port is located, and we did not have sufficient data in adverse weather conditions. However, we are currently collecting more data and plan to explore adverse situations in future research.
We appreciate your valuable feedback, which has greatly contributed to the enhancement of our work.
Round 2
Reviewer 1 Report
Thank you for revising the paper.
You addressed all my concerns. From scientific point of view, I see major work to be done to solve the limitations of your approach. However, the approach and work in progress is interesting.
Reviewer 3 Report
I acknowledge that it is not possible to carry out further checks on the data collected and therefore it is not currently possible to have certainty of the real effectiveness of the proposed procedures in an automated context. However, the scope of the research looks promising even if still under development.
I therefore leave the final judgment on the publication to the Editor.